# Lubricant Sensitivity of Direct Compression Grades of Lactose in Continuous and Batch Tableting Process

**DOI:** 10.3390/pharmaceutics15112575

**Published:** 2023-11-03

**Authors:** Gerald A. Hebbink, Pauline H. M. Janssen, Jurjen H. Kok, Lorenzo Menarini, Federica Giatti, Caterina Funaro, Salvatore Fabrizio Consoli, Bastiaan H. J. Dickhoff

**Affiliations:** 1DFE Pharma GmbH & Co. KG, 47574 Goch, Germanybastiaan.dickhoff@dfepharma.com (B.H.J.D.); 2Department of Pharmaceutical Technology and Biopharmacy, University of Groningen, 9713 AV Groningen, The Netherlands; 3IMA S.p.A. Active Division, 40064 Ozzano dell’Emilia Bologna, Italy; lorenzo.menarini@ima.it (L.M.);

**Keywords:** continuous manufacturing, excipients, lubrication, lubricant sensitivity, direct compression, lactose, content uniformity, blending

## Abstract

Modern pharmaceutical manufacturing based on Quality by Design and digitalisation is revolutionising the pharmaceutical industry. Continuous processes are promoted as they increase efficiency and improve quality control. Compared to batch blending, continuous blending is easier to scale and provides advantages for achieving blend homogeneity. One potential challenge of continuous blending is the risk of over-lubrication. In this study, blending homogeneity and lubricant sensitivity are investigated for both batch and continuous processes. Given their distinct chemical structures and morphologies, anhydrous lactose and granulated lactose are expected to exhibit varying sensitivities to changes in process settings across both technologies. The findings suggest that both lactose grades provide highly stable blends that can be safely utilised in both batch and continuous modes. Optimisation should focus on process variables, such as the quality of loss-in-weight feeders used for dosing low doses of ingredients. The most significant process parameter for lubricant sensitivity was the type of lactose used. Anhydrous lactose produced harder tablets than the more porous granulated lactose but was more sensitive to lubrication at the same settings. The magnesium stearate content and its interaction with the type of lactose are also critical factors, with magnesium stearate having a counterproductive impact on tabletability.

## 1. Introduction

Modern pharmaceutical manufacturing is revolutionising production processes in the pharmaceutical industry, with a focus on continuous processes [1,2]. This shift has forced the industry to redesign batch-wise processing into integrated continuous processing. Continuous processes offer significant advantages in producing tablets, as they enhance throughput and increase the robustness of production. Additionally, process control tools, such as process analytical technology (PAT), enable the detailed monitoring of processes [3,4,5].

Tablets are one of the most common pharmaceutical dosage forms, and their production requires the use of many ingredients. In addition to active pharmaceutical ingredients (APIs), excipients are used to facilitate the tableting process. Fillers enable tablet formation, and lubricants facilitate the tableting process. To obtain a blend suitable for compression from a process perspective, APIs are mixed with appropriate excipients that provide essential features, such as acceptable flowability and adequate compressibility.

Traditionally, pharmaceutical powder blending is performed using batch technology. Batch blending involves loading powders into large vessels (e.g., bins) that are tumbled for a fixed number of revolutions before the material is discharged and processed further. Although this technology has been extensively studied and explored, it presents some criticalities. Batch blending provides difficulty in scaling up, limited flexibility in batch sizes, and challenges for batch reproducibility. One of the major reasons for selecting continuous processes is to address these issues.

Continuous processes might, however, be difficult to design, as the existing knowledge on batch processes is not directly transferable to continuous processes. In order to transform batch production into continuous production, different manufacturing steps need to be re-designed. Multiple studies have compared the effect of raw material properties in batch and continuous processes. The research on continuous processes is increasing annually, focusing on the effects of process parameters, tablet characteristics, and formulation choices [6,7,8,9]. Additionally, research has been focused on the role of raw materials in different operating units, like feeders [10,11,12,13,14] and blenders [15,16,17,18,19,20,21,22]. Several studies have demonstrated that it is challenging to homogeneously mix materials with major differences in the particle size distribution (PSD) in standard batch processes, potentially causing segregation [23]. Since the scale of blending and the residence time in continuous manufacturing is significantly shortened, other studies have shown that this has a positive effect on blend homogeneity. So far, no studies have been identified that compare the impact of material properties in batch and continuous blending processes on the risks of over-lubrication.

When blending in a continuous direct compression (CDC) line, there could be a risk of over-lubricating the formulation. Excipients play a vital role in mitigating this risk. Excipients are required to be consistent; they need to have a proper powder flow to produce stable blends with other ingredients, and they should be compactable [24]. The correct use of lubricants in formulations is challenging, particularly for the development of formulations suitable for a CDC. Lubricants are required to eject tablets from dies without defects, but the inter-particle bonds lose strength owing to the potential formation of a thin layer of lubricant that covers APIs and excipient particles.

In this study, we investigated the role of anhydrous lactose and granulated lactose in both continuous and batch processes. These two types of lactose are ideal for both batch and continuous manufacturing because of their excellent powder flow and compactibility. Granulated lactose is composed of α-lactose monohydrate, whereas anhydrous lactose mainly consists of anhydrous β-lactose (80% w/w), with the remainder being anhydrous α-lactose [25,26,27,28]. Therefore, it is expected that these materials will exhibit different sensitivities to changes in the process settings for both technologies. Batch blending was performed using a tumble blender, whose design is characterised by an optimised geometry that ensures high mixing efficiency with no dead zones. Continuous blending was performed by a horizontal blender. The powder ingredients were continuously dosed directly into the blender by means of three different loss-in-weight feeders (LIW).

A process can become robust as soon as the correlations between critical powder attributes and the critical parameters of the process are well understood. The aim of this study is to better understand the requirements for processes and material properties in relation to lubricant sensitivity [29,30,31,32]. In a batch process, conditions such as mixing time, lubricant quantity, and mixing speed can be adjusted easily to account for variations in lubricant action. In a continuous process, the sensitivities to lubricants have not been deeply investigated yet, and further research is required to understand the interaction of process settings in combination with excipients. This is the first time that a quantitative comparison has been made between the lubricant sensitivity in batch and continuous blending.

## 2. Materials and Methods

### 2.1. Materials

A granulated lactose (SuperTab^®^ 30GR, DFE Pharma, Goch, Germany) and an anhydrous lactose (SuperTab^®^ 22AN, DFE Pharma, Goch, Germany) were blended with magnesium stearate (Ligamed MF-2-V, Peter Greven, Bad Muenstereifel, Germany) and FD&C blue #2, Indigo Carmine Aluminium Lake (Colorcon, Harleysville, PA, USA), as a model for an API. The tablets contained 1% w/w of the model API, 1–3% w/w of the lubricant, and the remaining 96–98% w/w was lactose.

### 2.2. Material Processing

#### 2.2.1. Batch Blending

The core of the batch setup was formed by a tumble-blender Cyclops MINI coupled with a 100 L bin (IMA S.p.A. Active, Ozzano dell’Emilia, Italy) with a diameter of 0.464 m. A pre-blend was prepared through bag mixing the API with part of the lactose. The pre-blend and the remaining lactose were blended at a constant speed of 15 rotations per minute (RPM) and a blending time of 15 min. The required quantity of magnesium stearate was added, and the second blending step was performed using the recipe parameters, as described in Section 2.5.

#### 2.2.2. Continuous Blending

Continuous blending was performed with the GCM450 horizontal continuous blender (Gericke, Regensdorf, Switzerland) with a diameter of 0.077 m. To continuously dose the powder, three different loss-in-weight feeders (LIW) were used. The largest feeder, responsible for dosing the lactose, had a hopper volume of 15 L. Two internal impellers ensured that the powder remained homogeneous before passing through the single screw. The model API was dosed using a LIW feeder with an 8 L hopper and double concave screws. The lubricant dosing LIW feeder had a 1 L hopper volume and double concave screws. The cumulative flow rates of the feeders were set to meet the required output of 45 kg/h. Before each design of experiments (DoE) run, a 15 min pre-blending step was conducted.

The blender had three separate inlet ports that allow the best residence time for each ingredient to be selected (Figure 1). The feeders for lactose and the model API were positioned at the first inlet. The lubricant inlet was positioned at the second or third inlet, depending on the DoE requirements. At the outlet of the blender, a weir was installed to enable the outlet to be partitioned, allowing the mean residence time of the blend to be modified. The material was continuously transported downstream from the continuous blender to the tablet press.

#### 2.2.3. In Process Analysis

A near-infrared (NIR) sensor (MicroNIR^®^ PAT-U, Viavi Solutions Inc., Scottsdale, AZ, USA) was installed as the process analytical technology (PAT) unit immediately before loading the powder into the tableting die. One sensor was located on the upper part of the lid of the bin of the batch blender, and one sensor was applied on the feed frame of the tablet die filling. The NIR sensor utilized a linear variable filter as a dispersive element, covering a wavelength range of 950 nm to 1650 nm. Raw spectra were pre-treated using a Standard Normal Variate and the first derivative via a Savitsky–Golay filter (11 pt, second polynomial order), before being analysed online using the moving block method with the F-Test. The wavelength range was selected to cover nearly the entire spectra, except for some lateral parts where the derivative pre-treatments create artificial noise. This qualitative method involved collecting independent data for comparing the variances of two blocks of data at a 95% confidence interval, enabling a real-time analysis of the material’s chemical composition and ensuring it remains under control throughout the entire run.

#### 2.2.4. Tableting

Tableting was performed with a Prexima 300 (Prexima 300, IMA Active, Ozzano dell’Emilia, Italy) with 9 mm round concave EU-B punches (iHolland, Nottingham, UK). Tablet press parameters were selected to ensure similar processes and results between the two grades of lactose investigated. A flat paddle was used for charging at 15 RPM. A flat paddle with increased volume was used for dosing at 40 RPM. A pre-compression force of 2 kN with a 1.5 mm upper punch penetration and a 3.5 mm pre-compression chamber was used. The main compression was performed with a 2.5 mm upper punch penetration and a compression chamber of 2.05 mm. The compression forces were 15.8 kN and 17 kN for anhydrous and granulated lactose, respectively. The tablet press turret speed was set to 60 RPM to achieve a throughput of 45 kg/h.

### 2.3. Powder Characterization

#### 2.3.1. Powder Density

The bulk and tapped densities (*n* = 2) were measured using Ph. Eur. Method 1. Specifically, 100 g of the powder was added to a 250 mL graduated cylinder, which was then placed on an automatic tapping device (STAV 2003 stampfvolumeter, Engelsmann, Ludwigshafen am Rhein, Germany). The Hausner ratio (HR) was calculated by dividing the tapped density (TD) by the bulk density (BD).

#### 2.3.2. Powder Flowability

A ring shear tester (RST-XS, Dietmar Schulze, Wolfenbüttel, Germany) was used to measure the flow function coefficient (ffc, *n* = 2). The flow function coefficient (ffc) was calculated as the consolidation stress divided by the unconfined yield strength. The powders were evaluated at a pre-consolidation stress of 4 kPa, whereas shear to failure was performed using normal stresses of 1, 2, and 3 kPa.

#### 2.3.3. Morphology

Scanning electron microscopy (SEM) images were obtained using a Phenom Pro 6 SEM (Thermo Fisher Scientific, Waltham, MA, USA). Samples were prepared by mounting a spoon tip of a powder blend on top of an SEM stub with the aid of double-adhesive conductive carbon tape. All samples were coated with 6 nm of gold with a LuxorAu sputter coater (Luxor Tech, Nazareth, Belgium) prior to SEM imaging. Images were collected at an acceleration voltage of 10 kV.

#### 2.3.4. Particle Size Distribution

Particle size distributions (*n* = 3) were determined using dry powder laser diffraction with a Helos/R instrument (Sympatec, Clausthal-Zellerfeld, Germany). The particle size distribution was reported as the volume-equivalent sphere diameter. A dry dispersion unit with a feed rate of 50% and air pressure of 0.5 bar was used. The particle size distribution was described via the cumulative undersize percentage, which is the percentage of particles smaller than a certain size. Specifically, the 10%, 50%, and 90% cumulative undersize values were denoted as d10, d50, and d90, respectively. Additionally, the span of the volumetric particle size distribution was calculated using the equation: Span = (d90 − d10)/d50. The span value indicates the width of the particle size distribution.

### 2.4. Tablet Analysis

During the trials, the tablets were monitored for weight, thickness, diameter, tensile strength (Charles Ischi AG, Zuchwil, Switzerland), model API content uniformity, and their relative standard deviations. The tablet tensile strength (*n* = 20, σ_t_) was calculated from the tablet crushing strength (*TCS*), the tablet diameter (*D*), and thickness (*t*) according to Equation (1):(1)$$\sigma_{t}=\frac{2\cdot TCS}{\pi \cdot D\cdot t}\;$$

The tablet tensile strength (*σ_t_*) was normalized by dividing it by the corresponding main compaction force (CF) to remove the natural variability in the CF from the dataset.

The content uniformity was determined by dissolving tablets (*n* = 10) in a borate buffer solution at a pH of 9.2 and measuring the UV-VIS absorption at 612 nm against a calibration line with a Spectrometer UV-6300PC (VWR^®^, Radnor, PA, USA). The content uniformity of the tablets was calculated as the relative standard deviation of the model API content of ten tablets from the same batch.

### 2.5. Design of Experiment

A full factorial design of the experiment (DoE) was performed. The aim was to change the most critical mixing process parameters for both technologies to evaluate the correlation between the process parameters and investigate the impact on tablets. The factorial designs are set up as summarised in Table 1 (the continuous process) and Table 2 (the batch process). The determined responses were the tablet tensile strength (σt), the tablet model API content, the model API content uniformity of the tablet, and the powder blend.

The responses were analysed using analysis of variance (ANOVA) statistics with the assistance of Minitab 20.3 statistical software. This approach allowed the relationships between the factors and the individual response variables to be identified and evaluated. Minitab 20.3 was also used to create Pareto charts, which display the standardised effects in descending order of magnitude from largest to smallest, with these effects represented as absolute values. The chart also includes a reference line, indicating the statistically significant effects at a *p*-value threshold of 0.1. The created models were summarised using a transfer formula that outlines the effect of individual factors on the response. Additionally, the model summary table includes several goodness-of-fit parameters, including the standard deviation of the distance between the data values and the fit (S), the percentage of variation in the response explained by the model (R^2^), the adjusted R^2^ (R^2^_adj_), which adjusts for the number of predictors in the model relative to the number of observations, and the predicted R^2^ (R^2^_pred_), which estimates how well the model predicts observations. All R^2^ values range from 0 to 100%, with higher values indicating a better fit.

## 3. Results and Discussion

Firstly, the physical properties of the two lactose grades were determined to understand the blend behaviour during the tableting process. Secondly, the continuous process was compared to the batch process by analysing the flow rate, blending regime, NIR results, and model API content to confirm their comparability and to investigate the process parameters that most affect tablets’ features. Then, the sensitivity of each process to the lubricant was evaluated by comparing the tensile strength of the tablets obtained under different process parameters. By identifying and addressing the key factors that affect the quality of the final product, it can be ensured that all the processes are robust, reliable, and capable of producing tablets with consistent quality.

### 3.1. Powder Characterization

An overview of the particle size distribution, density, and flow parameters of the lactose grades is summarised in Table 3. The SEM pictures are provided in Figure 2.

The particle size distributions of the lactose grades are relatively comparable, with anhydrous lactose being the coarsest of the two but with a tighter span than granulated lactose. In terms of powder flow, both excipients demonstrated excellent characteristics and are well suited for direct compression processes. Anhydrous lactose displayed a slightly higher density than granulated lactose. This distinction is particularly relevant in the pharmaceutical industry, where varying sizes of tableting dies are employed. The largest difference between the two powders was observed in the morphology. Anhydrous lactose is produced through the rapid drying of a lactose solution, followed by sieving to the desired particle size. Consequently, the formed particles are quite solid and have a smooth surface, with some irregularities that became visible only at higher magnifications [26]. Granulated lactose is an agglomerate of a fine-grade lactose that results in a structure consisting of an ensemble of fine particles bound together in a raspberry-like structure [33]. This results in a very irregular surface with numerous pores and holes.

### 3.2. Comparison of the Batch Versus the Continuous Process

To ensure that the mixing processes can be compared across different technologies (batch and continuous), a constant nominal flow rate value is maintained for all the trials at the tablet press level. The recipe parameters are also kept constant, apart from differences triggered by the difference in lactose type. This approach ensures that all the batches are produced under the same conditions, facilitating a comparison of the upstream process parameters. The Froude number, a dimensionless parameter that compares centrifugal and gravitational forces, is used to quantify the flow dynamics of the particles and determine the optimal blending regime for the specific technology being used. The Froude number (*Fr*) can be calculated from the tip speed (*v_tip_*), the equivalent radius (*r*), the gravitational acceleration constant (*g* = 9.81 m/s^2^), and the rotational speed (*RPM*):(2)$$Fr=\frac{v_{tip}^{2}}{r*g},\;\mathrm{with}$$
(3)$$v_{tip}=2*\pi *r*\frac{RPM}{60}\;.$$

Froude numbers were calculated for the range of DoE settings, as described in Table 2 and Table 3. The Froude numbers for the batch blending process varied from 0.05 to 0.21, all covering the gravity regime (*Fr* ≪ 1) of mixing [34]. Mixing in the gravity flow regime is characteristic for the tumbling blending method. Froude numbers for the continuous blending process varied from 0.3 to 2.8, covering different flow regimes. The lowest Froude numbers indicated a gravity flow regime (*Fr* ≪ 1), followed by a shear flow regime (*Fr* < 1) towards the centrifugal regime (*Fr* ≫ 1) [34]. This indicates that these two blending processes are inherently different in character. The main driver for the different flow regimes during batch and continuous blending is related to the different rotational speeds.

### 3.3. Process Stability

Process stability is required to obtain results that can be used to draw relevant conclusions. PAT control is supportive of evaluating the blend homogeneity that will be confirmed with tablet content uniformity through a chemical analysis in order to prove process stability.

#### 3.3.1. Uniformity of Powder Blends by NIR

To assess the stability of the process, an NIR probe was installed within the feed frame of the Prexima 300 tablet press, positioned just before the powder inlet. A moving block analysis was the method applied to monitor the process stability during the production of each batch. The trends displayed that only small deviations in the blend composition were present, which can be considered to be random noise. The standard deviations of the group of spectra were close to 0 (spikes of maximum 0.006%). Additionally, the mean absorbance remained constant, with a maximum discrepancy below 0.01. Therefore, the intra-batch homogeneity of the blends flowing through the tablet press can be considered sufficient [35].

#### 3.3.2. Tablet Content Uniformity

The model API content of the tablets manufactured is summarised in Figure 3, where the values are the averages of the ten tablets taken at three time points during each run of the whole DoE. The content uniformity (CU) was measured as the relative standard deviation (RSD) of the content of ten tablets taken at one time point and is shown in Figure 4.

It can be concluded that the CU is excellent in all cases, with average results of less than 2%. Although the CU is well within the requirements of a low-dose API [36], some minor differentiation can be observed. In the batch process, the granulated lactose performed slightly better than the anhydrous lactose. This is related to its morphology, i.e., more cavities and pores that can hold the low-dose fine model API. In the continuous process, in contrast, the anhydrous lactose performed slightly better than the granulated lactose, which is explained by the higher flowability of the anhydrous lactose.

An analysis of the CU results based on the DoE indicated no significant (α = 0.1) correlation between the CU and DoE factors. Both the lactose grades provided very stable blends and can, therefore, be safely used in both batch and continuous modes. Further optimisation could be performed, especially in this application where loss-in-weight feeders are used for dosing low amounts of ingredients.

### 3.4. Lubricant Sensitivity

The impact of the processing factors on the lubricant sensitivity of the tablet tensile strength (σ_t_) was studied for both blending modes. To analyse the DoE, only single-factor and first-order interaction terms were considered. Insignificant factors and interactions (*p* > 0.1) were eliminated from the model using a backward elimination approach. All the individual terms were included in the model when the interaction terms were significant.

#### 3.4.1. Continuous Mode

The significance of the impact of the process parameters on the tablet tensile strength is shown in the Pareto diagram in Figure 5. The transfer function of the normalised tablet tensile strength (σ_t_) for the continuous blender is provided in Equation (4):σ_t,continuous_ = 0.20778 − 1.906 B − 0.000001 C − 0.05572 D − 0.003354 E − 0.01065 F + 0.790 B∙D + 0.303 B∙F + 0.000051 C∙D + 0.003671 D∙E + 0.002713 D∙F − 0.003027 E∙F (4)

The type of lactose used was found to be the most significant process parameter, as each material has a specific impact on the tablet tensile strength (σ_t_). The anhydrous lactose was shown to produce harder tablets than the granulated lactose under the same tableting conditions. This is explained by the rough surface structure with clusters of microcrystals for the anhydrous lactose [37], which results in a higher degree of fragmentation [38,39]. The magnesium stearate content and its interaction with the lactose type were the second and third most crucial factors, respectively. It is well-known that magnesium stearate can have a negative effect on the tabletability of excipients [40]. The material-specific difference in sensitivity to magnesium stearate is responsible for the significant interaction term between the magnesium stearate content and lactose type. Anhydrous lactose has a higher sensitivity to magnesium stearate due to the flat morphology of this material. Blending for an extended number of revolutions causes the magnesium stearate to smear over this flat surface. In contrast, granulated lactose is less sensitive to over-lubrication due to its porous morphology. Magnesium stearate can smear less efficiently over the surface of this lactose grade, as magnesium stearate in the cavities of granulated lactose is shielded from blending shear forces.

The inlet port of the magnesium stearate was also a significant factor. An earlier introduction resulted in longer and more intense blending, with a negative impact on the tablet tensile strength. This also explains the significance of the interaction terms inlet port–lactose type and inlet port–magnesium stearate content. The weir type and the interaction terms weir type–lactose type and weir type–magnesium stearate were also significant. This was explained by the shape of the weirs. Weir type 2 has a full circular gap with no holdup of material before dropping it from the blender, while weir type 1 has a partial gap. Weir type 1 therefore holds some material, causing an increase in residence time and increased blending time with magnesium stearate.

The blending speed and the tableting moment during the process did not significantly affect the tablet tensile strength. The blending speed was not a significant factor, which was counterintuitive. An increased blending speed, however, corresponds to a reduced residence time in the blender, keeping the total blending energy more or less constant. The tablet production time was not significant either. The absence of a significant correlation between these parameters confirms the robustness of the blends and the process, as well as the stability provided by the excipients used.

#### 3.4.2. Batch Mode

The significance of the impact of the process parameters on the tablet tensile strength is shown in the Pareto diagram in Figure 6. The transfer function of the normalised tablet tensile strength (σ_t_) for the batch blender is provided in Equation (5):σ_t,batch_ = 0.19044 − 2.4184 B + 0.000161 C − 0.02856 D − 0.000177 E + 0.9106 B∙D − 0.000126 C∙D + 0.000063 D∙E(5)

The most dominant factor for the tensile strength (σ_t_) was the magnesium stearate content, followed by the type of lactose, and the number of revolutions. The magnesium stearate is a well-known factor that was shown to have a negative effect on the tabletability of the excipients for both blending processes [40]. The difference in morphology between the two materials also was a crucial factor. Magnesium stearate delaminates easier on the flat surface of anhydrous lactose than on the porous surface of granulated lactose, as discussed in Section 3.4.1. The number of revolutions was negatively correlated with the tablet tensile strength. The number of revolutions is a measure of the blending energy, and a higher number of revolutions results in more stress on the magnesium stearate to smear over the surface of the lactose. The filling degree was positively correlated with the tensile strength, although this impact was only minor. The filling degree is a well-known factor that can influence the mixing efficiency. This correlation is explained by the small reduction in mixing efficiency with a high filling degree [41,42]. With a high filling degree, the mixer has less space to accommodate bed dilation, which could have a negative impact. The moment at which the tablets were made during the process was not a significant factor and therefore was removed from the statistical model.

#### 3.4.3. Comparison of Continuous and Batch Modes

Figure 7 shows contour plots of the normalised tablet tensile strength for the batch and continuous mixing of anhydrous and granulated lactose. The normalised tablet tensile strength (norm. TTS) refers to the tablet tensile strength normalized by the compaction force and therefore has the unit MPa/kN. The plots indicate that tableting with anhydrous lactose resulted in harder tablets than tableting with granulated lactose. The tablet tensile strength of anhydrous lactose is however more sensitive to variation than the tablet tensile strength of granulated lactose.

The tablet tensile strength for >95% of the formulations was above 1.7 MPa. A tensile strength greater than 1.7 MPa will usually suffice to ensure that a tablet is mechanically strong enough to withstand commercial manufacturing and subsequent distribution. Tensile strengths down to 1 MPa may suffice for small batches where the tablets are not subjected to large mechanical stresses [43].

The formulations with the most extensive lubrication also had acceptable tablet tensile strengths of >1.5 MPa. This can be related to the brittle nature of lactose. During compaction, lactose particles break, which provides new surfaces for the bonding of the tablet [44]. The bonding capability of lactose particles is therefore not limited by the outer surface, which can be covered by the lubricant.

In order to calculate the sensitivity of the batch and continuous processes towards lubrication, the parameter magnesium stearate sensitivity (MSS) was introduced. This factor indicates the change in tablet tensile strength when the magnesium stearate concentration is changed and therefore indicates the risk for over-lubrication.

The magnesium stearate sensitivity (*MSS*) can be described as the first derivative of the tablet tensile strength (*σ_t_*) as a function of the magnesium stearate content (*MgSt*), as described in Equation (6):(6)$$MSS=\frac{d\;\sigma {}_{t}}{d\;MgSt}$$

Using the transfer function for continuous blending (Equation (4)), this results in an MSS for continuous blending that depends on the lactose type (*D*) and the magnesium stearate inlet port (*F*), as shown in Equation (7):(7)$${MSS}_{continuous}=-1.9+0.79\cdot D+0.303\cdot F\;\pm \;0.19$$

Using the transfer function for batch blending (Equation (5)), this results in an *MSS* for batch blending that depends only on the lactose type (Type), as shown in Equation (8):(8)$${MSS}_{batch}=-2.42+0.91\cdot D\;\pm \;0.11$$

Equations (7) and (8) provide the formulas that describe the sensitivity of the tablet tensile strength to variations in lubricant levels, which were derived from the transfer formulas from the Design of Experiments (DoEs). The *MSS* depends on different factors for a batch process than for a continuous process. Over the entire design space that is tested, the *MSS* depended mainly on the type of lactose grade used. The anhydrous lactose had a higher *MSS* than the granulated lactose. For a continuous process, the inlet port for the magnesium stearate was also important. A higher sensitivity for magnesium stearate was observed when the magnesium stearate was introduced at inlet port 2 compared to inlet port 3, which is explained by the longer blending time when inlet port 2 is used.

These formulas result in MSS values for anhydrous and granulated lactose, as displayed in Figure 8. The dependency of lubricant sensitivity in interaction with the type of lactose is of the same order for both processing modes. This shows that the MSS is a material property that is relatively independent of the equipment and blend method used.

Figure 8 shows that the magnesium stearate sensitivity of anhydrous lactose is approximately twice as big as the magnesium stearate sensitivity of granulated lactose. The higher sensitivity of anhydrous lactose to lubrication is explained by the morphology of the materials, as indicated in Section 3.4.1.

Figure 9 shows the surface of anhydrous lactose and granulated lactose after batch blending with 3% w/w magnesium stearate for 200 revolutions. Anhydrous lactose comprises solid particles with flat, rough surfaces. Magnesium stearate therefore increasingly spreads on the surface of these particles upon increased blending energy. Granulated lactose, in contrast, has a more open structure. The small magnesium stearate particles fill the cavities, providing protection against further spreading of magnesium stearate on the surface.

Even though magnesium stearate is partly captured in the cavities of granulated lactose, lubrication was still, as no picking or sticking was observed. This is in line with research, e.g., by Ragnarsson et al., who showed that a poor distribution of magnesium stearate over the blend does not necessarily hurt lubrication efficiency [45,46].

## 4. Conclusions

The stability and magnesium stearate sensitivity of the formulations with anhydrous lactose and granulated lactose were investigated in both batch and continuous processes. Both processes were found to be highly stable over time with excellent content uniformity, with average relative standard deviations less than 2%. In the batch process, the granulated lactose performed slightly better than the anhydrous lactose. This is related to its morphology, which has more cavities and pores. In contrast, in the continuous process, the anhydrous lactose preformed slightly better than the granulated lactose, which is explained by the higher flowability of anhydrous lactose.

The magnesium stearate sensitivity for the defined formulations was similar in the batch and continuous processes. The magnesium stearate sensitivity in the continuous process depended on the inlet port and the lactose type used, while for batch blending, the lactose type was the only relevant factor. Anhydrous lactose showed better compactibility, but granulated lactose had lower magnesium stearate sensitivity. A thorough understanding of these interactions is crucial for optimal and robust, continuous production processes.

## Figures and Tables

**Figure 1 pharmaceutics-15-02575-f001:**
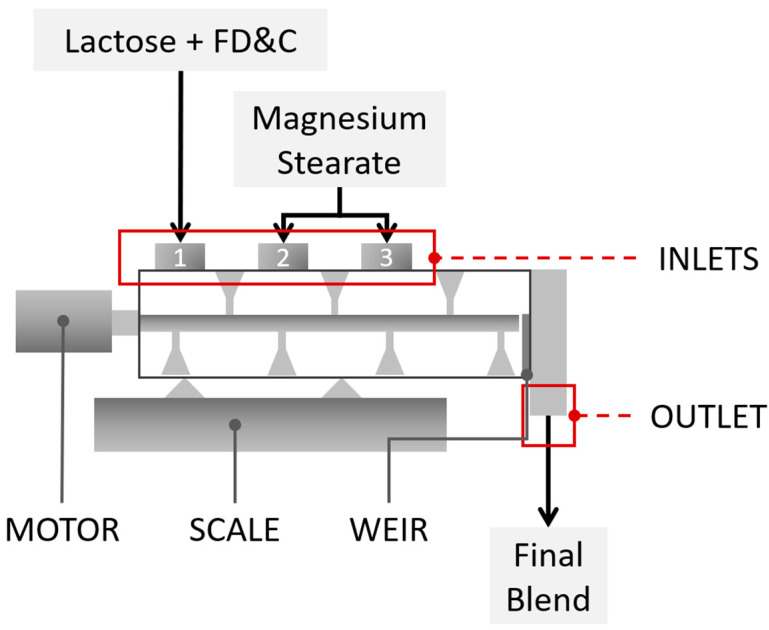
Continuous blending process scheme.

**Figure 2 pharmaceutics-15-02575-f002:**
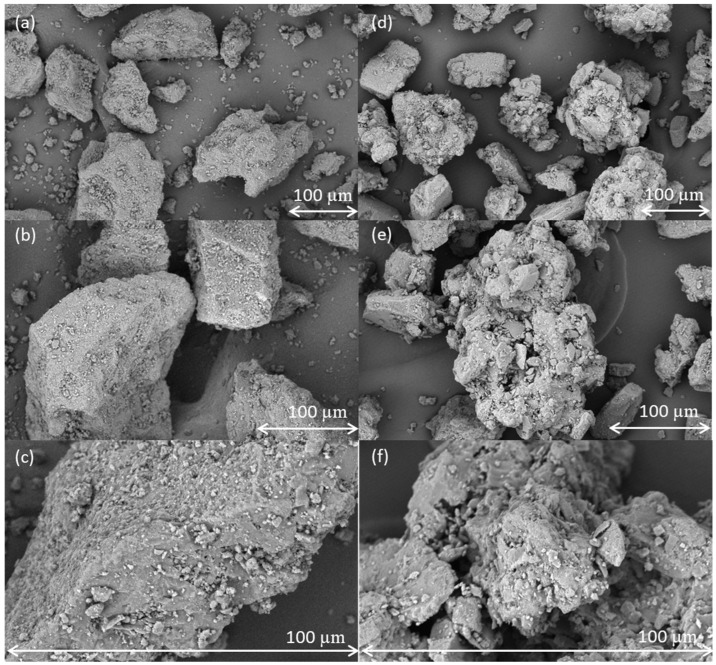
SEM images of anhydrous lactose (**a**–**c**) and granulated lactose (**d**–**f**) at magnifications of 1000× (**a**,**d**); 1500× (**b**,**e**) and 5000× (**c**,**f**).

**Figure 3 pharmaceutics-15-02575-f003:**
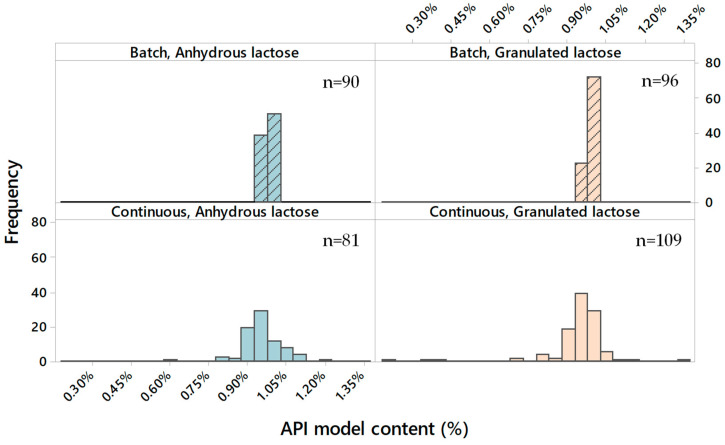
The average model API content remained consistent across all runs, with no discernible difference observed between the two types of lactose.

**Figure 4 pharmaceutics-15-02575-f004:**
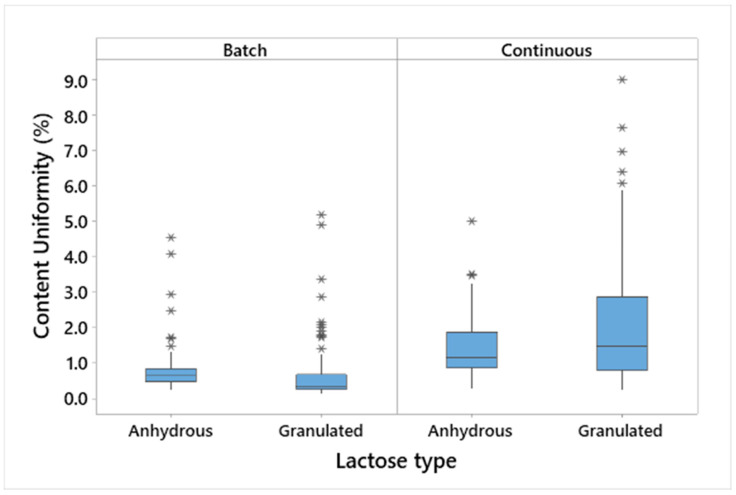
Boxplot showing the content uniformity (CU) of tablets. The middle line represents 50% of all data; the top and bottom of the box represent the 25% and 75% quartiles. Whiskers represent the minimum and maximum of the data, excluding outliers. Outliers are represented by a star (*).

**Figure 5 pharmaceutics-15-02575-f005:**
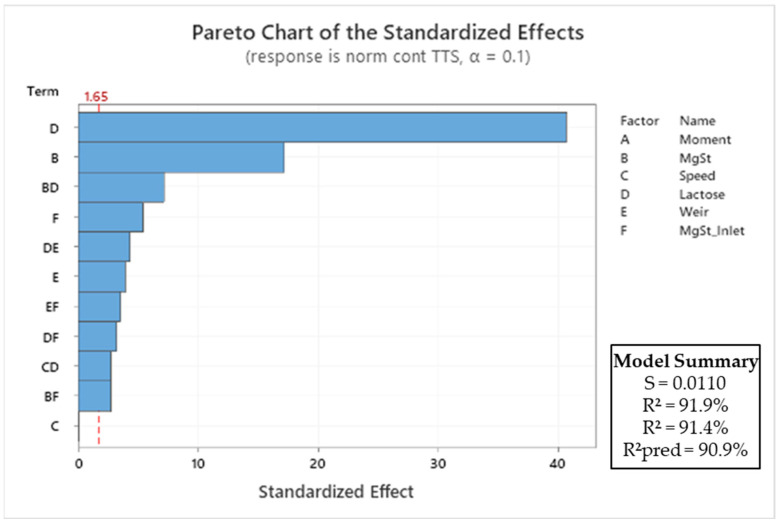
Pareto diagram of standardised effects of process parameters on tablet tensile strength (σ_t_) for the continuous process. All insignificant factors (*p* > 0.1) were removed from the model, and only first-order and interaction terms were considered.

**Figure 6 pharmaceutics-15-02575-f006:**
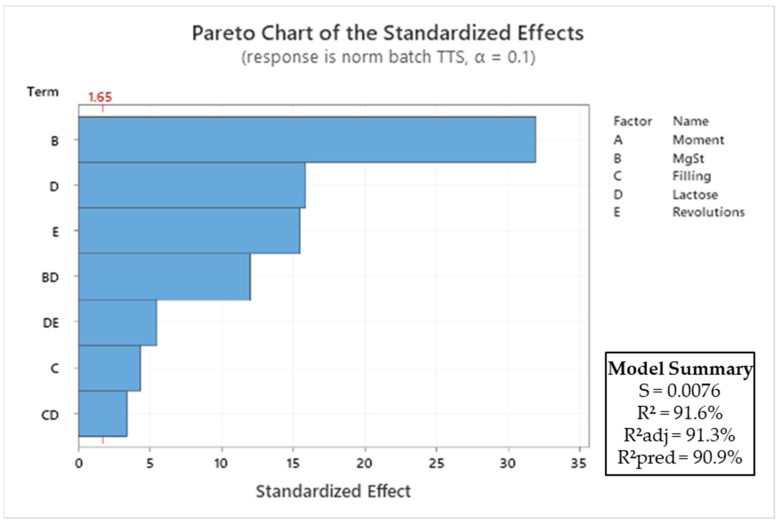
Pareto diagram of standardised effects of process parameters on tablet tensile strength (σ_t_) for the batch process. All insignificant factors (*p* > 0.1) were removed from the model, and only first-order and interaction terms were considered.

**Figure 7 pharmaceutics-15-02575-f007:**
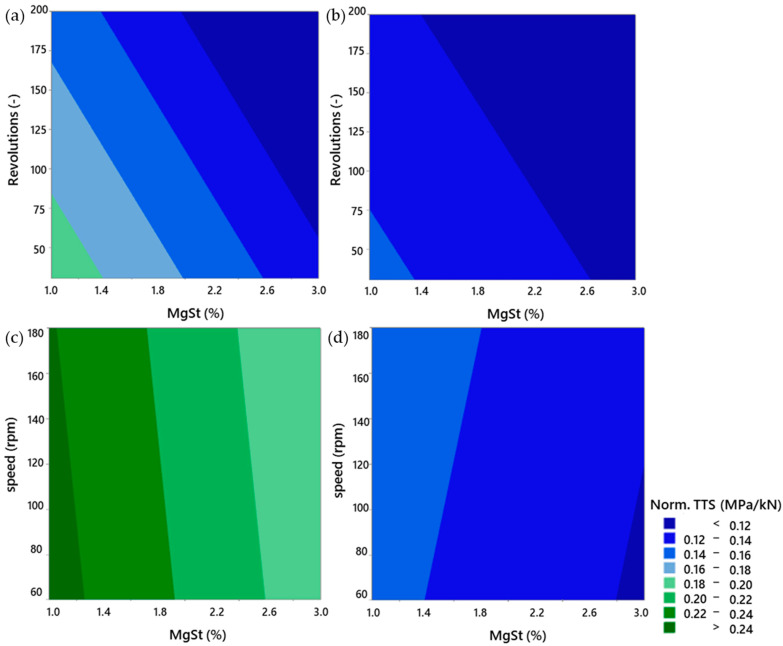
Contour plots of normalised tablet tensile strength for batch processes (**a**,**b**) and continuous processes (**c**,**d**) for formulations containing anhydrous lactose (**a**,**c**) and granulated lactose (**b**,**d**).

**Figure 8 pharmaceutics-15-02575-f008:**
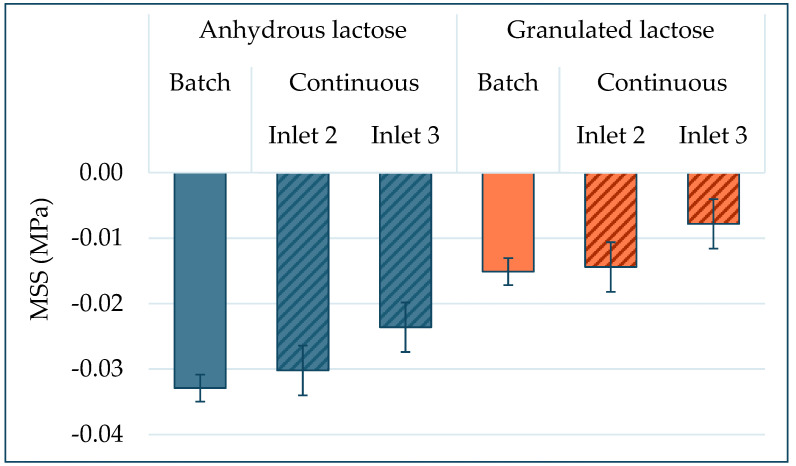
Magnesium stearate sensitivity (MSS) for anhydrous lactose and granulated lactose when processed with batch and continuous blending modes.

**Figure 9 pharmaceutics-15-02575-f009:**
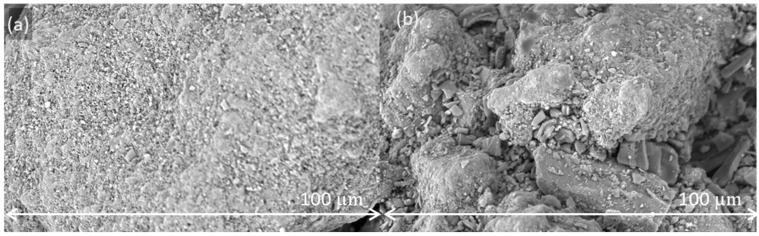
SEM images of blends with anhydrous lactose (**a**) and granulated lactose (**b**) blended with 3% w/w magnesium stearate and 200 revolutions in a batch blender at a magnification of 5000×.

**Table 1 pharmaceutics-15-02575-t001:** Factors used in the full factorial design of the experiment (DoE) in continuous mode blending.

Code	Factor	Coded	Factors	Type
A	Moment during tableting	Moment	Start, middle, end	Attribute
B	Magnesium stearate content (% w/w)	MgSt	1–3	Continuous
C	Blending tool speed (RPM)	Speed	60–180	Continuous
D	Lactose type	Lactose	Anhydrous lactose, Granulated lactose	Attribute
E	Weir type	Weir	1, 2	Attribute
F	Magnesium stearate inlet port	MgSt_Inlet	Port 2, Port 3	Attribute

**Table 2 pharmaceutics-15-02575-t002:** Factors used in the full factorial design of experiment (DoE) in batch mode blending.

Code	Factor	Coded	Factors	Type
A	Moment during tableting	Moment	Start, middle, end	Attribute
B	Magnesium stearate content (% w/w)	MgSt	1–3	Continuous
C	Percentage of bin filling (%)	Filling	30–70	Continuous
D	Lactose type	Lactose	Anhydrous lactose, Granulated lactose	Attribute
E	Number of tumbling revolutions	Revolutions	30–200	Continuous

**Table 3 pharmaceutics-15-02575-t003:** Particle size, density, and flow properties of anhydrous lactose and granulated lactose.

	Anhydrous Lactose	Granulated Lactose
d10 (µm)	47	38
d50 (µm)	203	126
d90 (µm)	359	297
Span (-)	1.54	2.05
FFC @ 4kN (-)	17	17
Bulk density (g∙cm^−3^)	0.68	0.55
Tapped density (g∙cm^−3^)	0.80	0.67
Hausner ratio (-)	1.17	1.22

## Data Availability

The data presented in this study are available on request from the corresponding author.

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
