# Peer review of "Lubricant Sensitivity of Direct Compression Grades of Lactose in Continuous and Batch Tableting Process"

_pharmaceutics, 2023, doi:10.3390/pharmaceutics15112575_

Round 1

Reviewer 1 Report

Comments and Suggestions for Authors

The work by Hebbink et al. explores the effects of batch and continuous type processes on the content uniformity and tablet tensile strength of two different directly compressible lactose grades, i.e., granulated and anhydrous lactose. It’s an industrially applicable research. However, the research work seems incomplete, with a few inconsistencies observed, as indicated below.

1.          Authors should include a few other studies where researchers have tried to modify processes for continuous manufacturing.

2.          Figure 4: There appears to be some error in the unit of content uniformity. Recheck and correct the same.

3.          Page 9, Line 292: “In the batch process, granulated lactose performed slightly better than anhydrous lactose. This is related to its morphology, i.e., more cavities and pores that can hold the low dose fine model API”.

The authors are suggested to provide evidence for the claims. Fluorescence microscopy can be performed to show that the model API is held in the cavities of granulated lactose. Also, mention why the granulated lactose did not perform well in the continuous process.

4.          Page 10, Line 326: “Blending for an extended number of revolutions causes the magnesium stearate to smear over this flat surface.”

The SEM image of anhydrous lactose does not show a flat surface. The particles exhibit a granular/rough surface. Also, there is no conclusive evidence shown to support the above claim.

5.          Page 11, Line 329: “Magnesium stearate can smear less efficiently over the surface of this lactose grade, as the magnesium stearate in the cavities of granulated lactose is shielded from blending shear force.”

Does the confinement of magnesium stearate in the cavities of granulated lactose have any negative effects on tablet lubrication?

6.          Figure 7: The units of tablet tensile strength should be mentioned, and the official pharmacopeial limits for tablet tensile strength should be mentioned.

7.          Figure 7 shows that the granulated lactose grade had poor tablet tensile strength in both batch and continuous process. However, no reason is mentioned for this observation.

8.          The magnesium stearate sensitivity (MSS) part of the manuscript is unclear and poorly written. Explain the importance of MSS and also explain why the values of the Y axis of Figure 8 indicate.

9.          Figure 8: The values are negative and are shown in the positive Y axis. Cross-check if the representation of the values in the graph is correct.

10.      Discussions seem to end abruptly by only providing the data. Authors should discuss the data in detail and try to correlate the results.

11.      At multiple locations, authors have mentioned the Figures and did not mention the figure number. 

Comments on the Quality of English Language

NA

Reviewer 2 Report

Comments and Suggestions for Authors

The authors compared batch-based and continuous blending methods in frame of their sensitivity to magnesium stearate and over-lubrication. The concept is interesting and may have high interest as the importance of continuous processes is emerging in field of pharmaceutical manufacturing.

The paper is generally well written, but some important information is missing, and therefore I am not fully convinced that the results fully support the conclusions. Beside of the morphology the surface free energy of the components and the binding sites on the surface of lactose is very important from the aspect of over-lubrication. SFE data of lactose samples and MgSt and/or SEM pictures of the blended samples would be advantageous to clarify the differences.

I would also debate with that conclusion that two lactose shows no differences in batch/continuous processes, as it is clearly visible on Figure 7 that the interaction of speed and MgSt has completely different importance in case of the two lactoses.

Displaying NIR spectra of blend homogeneity would be necessary.

Minor comments

Line 179 please write Switzerland instead of Switserland

I would be better the use of the coding of parameters for MSS equations too.

Reviewer 3 Report

Comments and Suggestions for Authors

The topic of lubrication is a really old topic, with over 40 years of work in batch blending. However, it is a topic that needs revisiting with the advent of continuous manufacturing and with variations in the excipents used. 

The following statement is not clear: “An optimised blending tool was used to blend and transport the materials simultaneously, establishing combined radial and axial mixing.” What was the tool used?

The NIR is mentioned, but the manuscript does not include a single spectrum. Was the entire NIR spectral region used in the moving block? The moving block could be focused on the API region, or it could be focused on the entire spectral region. 

Were any spectral differences observed when the lubricant was increased from 1 to 3% w/w? 

Round 2

Reviewer 1 Report

Comments and Suggestions for Authors

The authors have addressed all the concerns. 

Reviewer 2 Report

Comments and Suggestions for Authors

The authors properly answered my concerns.